# Lipophilic Bioactive Compounds Transported in Triglyceride-Rich Lipoproteins Modulate Microglial Inflammatory Response

**DOI:** 10.3390/ijms23147706

**Published:** 2022-07-12

**Authors:** Juan M. Espinosa, Jose M. Castellano, Silvia Garcia-Rodriguez, Angélica Quintero-Flórez, Natalia Carrasquilla, Javier S. Perona

**Affiliations:** Department of Food and Health, Instituto de la Grasa-CSIC, Campus of the University Pablo de Olavide, Building 46, 41012 Seville, Spain; spinnosa@gmail.com (J.M.E.); jmcas@ig.csic.es (J.M.C.); silvia11_12@hotmail.com (S.G.-R.); angelica.qf@gmail.com (A.Q.-F.); n.carrasquilla.garcia@gmail.com (N.C.)

**Keywords:** bioactive compounds, inflammation, microglia, oxidative stress, triglyceride-rich lipoprotein

## Abstract

Microglial cells can contribute to Alzheimer’s disease by triggering an inflammatory response that leads to neuronal death. In addition, the presence of amyloid-β in the brain is consistent with alterations in the blood–brain barrier integrity and triglyceride-rich lipoproteins (TRL) permeation. In the present work, we used lab-made TRL as carriers of lipophilic bioactive compounds that are commonly present in dietary oils, namely oleanolic acid (OA), α-tocopherol (AT) and β-sitosterol (BS), to assess their ability to modulate the inflammatory response of microglial BV-2 cells. We show that treatment with lab-made TRL increases the release and gene-expression of IL-1β, IL-6, and TNF-α, as well as NO and iNOS in microglia. On the other hand, TRL revealed bioactive compounds α-tocopherol and β-sitosterol as suitable carriers for oleanolic acid. The inclusion of these biomolecules in TRL reduced the release of proinflammatory cytokines. The inclusion of these biomolecules in TRL reduced the release of proinflammatory cytokines. AT reduced IL-6 release by 72%, OA reduced TNF-α release by approximately 50%, and all three biomolecules together (M) reduced IL-1β release by 35% and TNF-α release by more than 70%. In addition, NO generation was reduced, with the inclusion of OA by 45%, BS by 80% and the presence of M by 88%. Finally, a recovery of the basal glutathione content was observed with the inclusion of OA and M in the TRL. Our results open the way to exploiting the neuro-pharmacological potential of these lipophilic bioactive compounds through their delivery to the brain as part of TRL.

## 1. Introduction

Alzheimer’s disease (AD) is the most common cause of dementia, and its prevalence is expected to quadruple by 2050. AD is characterized by neuronal and synaptic losses, chronic inflammation, and a decrease in cognitive abilities [1], associated with abnormal deposits of amyloid-β (Aβ) polypeptide and neurofibrillary aggregates of hyper-phosphorylated tau protein in the cerebral cortex [2]. Clinical trials and cross-sectional studies have shown a positive association between AD and atherosclerosis [3]. Common risk factors for these pathologies include hypertension, hypercholesterolemia, a sedentary lifestyle and a non-healthy diet [4]. In this regard, there is accumulating evidence supporting the hypothesis that certain lipids contribute to the onset and progression of AD by inducing and aggravating dyslipidaemia, endothelial dysfunction, inflammation and oxidative stress [5,6].

Dietary fats are transported in blood as chylomicrons, which are triglyceride-rich lipoproteins (TRL) containing Apolipoprotein (Apo) B48, secreted by enterocytes. A significant part of plasma Aβ is associated with TRL particles [7,8] and immunoreactivity to apolipoprotein (apo) B, a structural protein of TRL, has been found in the postmortem brain parenchyma of subjects with AD [9]. In addition, it has been consistently suggested that postprandial dietary fats have an impact on TRL-Aβ secretion and that repeated postprandial excursions may disrupt blood-brain barrier function, raising the entry of toxins, pathogens, and molecules, such as lipoproteins, into the brain [10]. Studies in murine models have shown that chronic ingestion of an SFA-rich diet significantly increases the Aβ synthesis by enterocytes, generating an exaggerated postprandial response in enriching the Aβ content in TRL, compared to a low-fat or MUFA- and PUFA-rich diets [11].

Microglia are significant contributors to the immune response within the central nervous system. Similar to the periphery macrophages, microglia adapt their phenotype in response to microenvironmental insults, expressing both a proinflammatory (‘activated’) phenotype and an alternative anti-inflammatory (‘non-activated’ or ‘resting’) one [12]. In the proinflammatory state, microglia release proinflammatory cytokines and reactive oxygen and nitrogen species [10,13]. The overactivation of microglia has been associated with pathophysiological processes related to AD.

It has been recently reported that postprandial chylomicrons polarize BV-2 microglial cells towards the proinflammatory phenotype, promoting the gene expression and release of TNF-α, IL-1β and IL-6 cytokines [14]. These effects would be triggered by the Apo B48 receptor (ApoB48R) transcriptional activity. Microglia activation through ApoB48R upregulation succeeds in an FA-dependent manner, being more pronounced with saturated fatty acids-rich TRL (TRL-SFA) in comparison with those rich in monounsaturated and polyunsaturated fatty acids (TRL-MUFA or TRL-PUFA) [15].

Given that sustained postprandial hypertriglyceridemia can increase susceptibility to diseases related to neuroinflammation, we hypothesize that an interesting nutritional strategy could be the incorporation into the diet of foods or dietary preparations rich in lipophilic bioactive compounds with anti-inflammatory and antioxidant activity, which would be encapsulated in chylomicrons together with the rest of the dietary fat, and would exert an inhibitory action on the proinflammatory phenotype of microglia cells.

Phytosterols, tocopherols and triterpenoids present in plant foods and medicinal herbs, have relevant biological activities at rather low concentrations. Evidence suggests that they can interact with different metabolic pathways, exerting beneficial effects against a number of chronic degenerative pathologies associated with obesity, insulin resistance and metabolic syndrome, including neurodegenerative diseases. They may modify the expression of key genes in the adaptive cellular response, mitigating oxidative and chemotoxic stresses and inflammation [16].

We have previously reported that pretreatment with oleanolic acid (OA, 3b-hydroxy-olean-12-en-28-oic acid), a natural pentacyclic triterpene, causes a reduction in the release of inflammatory mediators in LPS-stimulated BV-2 microglial cells [17]. OA is significant in the olive tree (*Olea europaea* L.) and its products, including virgin and pomace olive oils [18,19]. These oils are also rich in α-tocopherol (AT) [20] and β-sitosterol (BS) [21]. There is evidence that AT and BS are transported in blood as components of TRL during the postprandial period [22]. We theorize that OA, due to its strictly lipophilic nature, could also be systemically transported within TRL.

Chylomicrons and their remnants are difficult to isolate from human blood due to the concomitant presence of other lipoproteins during fractionation by density gradient. For that reason, laboratory-made artificial TRL has been used as models to study their metabolism as they mimic the effects of real chylomicrons in different cell models and provide better reproducibility [23,24]. In addition, the method allows reliable incorporation of lipophilic molecules at controlled concentrations [25]. In this work, we used lab-made TRL as carriers of OA, AT and BS (TRL-OA, TRL-AT and TRL-BS, respectively) to assess their ability to interact with BV-2 cells and modulate their inflammatory response.

## 2. Results

### 2.1. Protein and Gene Expression of Proinflammatory Cytokines in BV-2 Cells

The internatization of TRL into BV-2 cells was verified by the presence of intracellular lipids stained with Oil Red O and observed in mucriophotographs (Figure 1). The release of inflammatory cytokines was measured using the ELISA method. It was observed that treatment of BV-2 cells with lab-made TRL (positive control) resulted in enhanced release of proinflammatory mediators compared with non-treated cells, doubling the concentration of the inflammatory cytokines IL-1β, IL-6 and TNF-α in the culture media (Figure 2).

This response was attenuated when TRL included the bioactive compounds in their composition. The effect of TRL carrying individual bioactive compounds was modest when IL-1β was analysed, with slight non-significant reductions of this cytokine when compared with the treatment of lab-made TRL (Figure 2a). However, the concurrent presence of OA, AT and BS in the TRL significantly reduced IL-1β production.

The release of IL-6 by BV-2 cells was reduced by 23% on average due to the treatment with TRL-OA compared to the treatment with the lab-made TRL devoid of bioactive compounds. However, this reduction did not achieve statistical significance (Figure 2b). On the other hand, BS did not affect the IL-6 release. The most significant effect was obtained through treatment of the microglial cells with TRL containing AT, with a 72% reduction compared to the control TRL. This level of IL-6 production represented half of that of the non-treated cells. Surprisingly, the incubation of BV-2 with TRL holding the three bioactive compounds (TRL-M) did not affect the IL-6 release compared to treatment with lab-made TRL.

Regarding TNF-α, the presence of OA in TRL strongly diminished the release of this cytokine to the culture medium by more than a half when compared with the incubation with TRL without the triterpene (Figure 2c). TRL, including AT or BS, only caused a very slight decline in the TNF-α production in BV-2 cells, which was not significant. Nevertheless, the inclusion of the three bioactive compounds in the composition of the TRL reduced TNF-α production by more than 70%.

To further study the mechanisms underlying cytokine release, we measured the gene expression of these mediators in TRL-stimulated BV2 cells by reverse transcription-quantitative polymerase chain reaction (RT-qPCR). The burst in IL-1β, IL-6 and TNF-α caused by lab-made TRL was due, at least partially, to an augmented de novo synthesis of these proteins since the expression of genes encoding for them experienced significant increases: 2.3-fold for TNF-α, 6.6-fold for IL-1β and 10.0-fold for IL-6.

When cells were treated with TRL containing the bioactive compounds, the expression of those genes was reduced at different extensions. IL-1β gene expression was strongly depressed (c.a. 50%) by AT or BS containing TRL and TRL-M (Figure 3a). However, the effect was even more severe when cells were incubated with TRL-OA since the expression of IL-1β was reduced by near 95%, down to levels even lower than those obtained in non-treated cells. TRL-OA also showed the strongest down-regulatory effect on IL-6, reducing the expression of its gene by 95% (Figure 3b). Likewise, the incubation of BV-2 cells with TRL-AT caused a significant reduction (51%) of this gene compared to the treatment with lab-made TRL lacking bioactive compounds. In the case of IL-1β, the TRL-BS treatment seemed to reduce the gene expression by 15–17%, although this decline did not achieve statistical significance. Unexpectedly, the treatment of BV-2 cells with TRL-M did not have a significant effect on the production of RNA transcripts.

A similar lowering effect on mRNA expression was found for TNF-α when microglial cells were incubated with TRL-OA and TRL- AT (Figure 3c). In this case, the influence of TRL-AT was even more remarkable. Concerning the behaviour of BV-2 cells when treated with TRL-BS, the inclusion of the phytosterol failed to repress the expression of this cytokine. In contrast to the IL-6 gene, incubation of cells with TRL-M caused powerful repression of the TNF-α gene expression, likely because of the individual contributions of OA and AT.

### 2.2. NO Release and Expression of iNOS in BV-2 Cells

We also investigated the effect of OA, AT and BS, as part of TRL, on NO production. Stimulation of BV-2 cells with lab-made TRL (positive control) doubled NO release compared to saline (Figure 4a). Cell incubation with TRL-OA and TRL-BS prevented this rise, and NO production was repressed by 45% (TRL-OA) and 80% (TRL-BS). TRL-AT caused the opposite effect, and a significant rise of nitrite content above the values obtained with the lab-made TRL-treated cells was registered. In contrast, NO production in BV-2 cells treated with TRL-M was strongly depressed, with an 88% reduction. This effect should be derived from the additive contributions by TRL-OA and TRL-BS.

NO production is regulated by the activity of inducible NO synthase (iNOS). Therefore in this work, we also analysed how its gene expression is affected by the treatments with the different TRL. The incubation of BV-2 cells with control lab-made TRL exacerbated the expression of iNOS by 7-fold (Figure 4b). In contrast, treating BV-2 cells with TRL-OA caused a powerful (95%) repression of the gene expression. The inclusion of AT and BS in TRL only caused a very modest decline in iNOS gene expression, which was not statistically significant.

### 2.3. Glutathione Production in BV2 Cells

The incubation with lab-made TRL doubled the intracellular concentrations of total glutathione (GSH + GSSG) in BV-2 cells in comparison with that in the untreated cells (Figure 5). Among the bioactive compounds studied, the inclusion of OA in the TRL was the only one capable of restoring GSH to baseline level. In cells treated with TRL-AT and TRL-BS, GSH contents remained high, although somewhat lower than in lab-made TRL-treated cells. In addition, when the cells were incubated with TRL-M, a similar effect to that found for the treatment with TRL-OA was observed, which suggests that this lowering effect should be attributed to OA almost exclusively.

## 3. Discussion

Microglial cells are brain-resident macrophages that participate in controlled immune reactions that are essential for neuron survival [26,27]. However, in pathophysiological conditions, the exposure to misfolded proteins (such as Aβ or phosphorylated tau) and neuronal damage signals over-activate microglia and triggers the production of free radicals and proinflammatory cytokines, generating a positive feedback loop between neuronal injury and inflammation which causes neuronal death and the activation of other glial cells [1,28]. Therefore, modulation of microglial function is a therapeutic target for neurodegenerative diseases [29].

Animal studies have shown that high-fat and high-cholesterol diets exacerbate amyloid accumulation, increase neuroinflammation, disrupt BBB integrity and induce cognitive impairment [30,31,32,33]. Lipolytic products of TRL transiently elevate BBB permeability and induce lipid droplet formation in astrocytes, activate cellular stress pathways and increase secretion of inflammatory cytokines [34].

In the present work, we report the effect of lab-made TRL, which mimic human postprandial TRL, on BV-2 cell release of radical species and inflammatory mediators. TRL were able to increase the release and gene expression of IL-1β, IL-6 and TNF-α. In this regard, previous research by Toscano et al. [15] determined that human postprandial TRL could modulate microglial plasticity depending on different dietary fatty acids, with SFA-enriched-TRL enhancing the proinflammatory microglial phenotype.

As stated above, the inflammatory response of microglial cells also includes an acute upsurge of NO production, governed by the over-expression of iNOS [35]. In this work, treatment of BV-2 cells with lab-made TRL doubled the release of NO and increased the iNOS expression by 7-fold. Microglia has a battery of antioxidant agents to maintain redox homeostasis in the brain under both normal physiological and pathophysiological conditions. As part of this antioxidant defence, microglial cells regulate the levels of glutathione (γ-glutamylcysteinyl glycine; GSH) [36]. GSH is the most abundant non-protein thiol in all kinds of cells. Its functions are diverse, including the maintenance of the intracellular antioxidant system and redox homeostasis. All of its major biological processes involve the redox balance of the thiol residue within the molecule. Two GSH molecules are oxidized to produce one molecule of GSH disulphide (GSSG) in order to eliminate ROS/RNS, and GSSG can be reduced back to two GSH molecules via reaction with GSH reductase (GR) [37].

In microglia, GSH synthesis seems to be focused on eliminating ROS generated under pathological conditions. GSH levels and their related enzyme activities, such as glutathione peroxidase (GPx) and GR, are higher in cultured microglia, especially under oxidative stress conditions [38,39]. Since, the microglial cells themselves produce superoxide radicals and NO when activated for the phagocytosis of dead cells, foreign molecules and pathogens, they must have a sufficient defense mechanism against oxidative stress [40,41].

Activated microglia can also express system xc(-) [42] and the glutamate transporter-1 (GLT-1) [43]. The first can exchange extracellular cystine for intracellular glutamate and then intracellularly reduce cystine back to two Cys molecules that can be utilized as a substrate for GSH synthesis, whereas the second promotes the reuptake of the excreted glutamate by system xc- for use in GSH synthesis [44]. Since the microglia are exposed to large amounts of RON/RNS, especially under pathological conditions, the coupling between the system xc- and GLT-1 plays a critical role in microglial GSH synthesis against oxidative stress.

Our research shows that treatment of BV-2 cells with TRL doubles the intracellular pool of glutathione (GSH + GSSG) when compared to untreated cells. Altogether, our results provide new evidence supporting that TRL activate microglial cells and therefore could play a role in neuroinflammation.

On the other hand, our research also offers novel data indicating that these lipoproteins could be carriers for lipophilic bioactive compounds (OA, AT and BS) with antioxidant and anti-inflammatory effects, which might attenuate the release of proinflammatory mediators by microglial cells. The inclusion of OA within lab-made TRL significantly reduced IL-6 and TNF-α release by BV-2 cells, and strongly repressed IL-1β, IL-6 and TNF-α gene expression. In addition, the presence of the triterpene in the TRL core powerfully inhibited the expression and the activity of iNOS, thus restraining the generation of NO.

These results are consistent with those previously reported by us, revealing that the treatment of BV-2 cells with OA (in DMSO) attenuated the inflammatory response elicited by the exposure to LPS [17]. Compared to LPS, which is commonly used to stimulate the inflammatory response in different cell types, stimulation of BV-2 cells with TRL led to lower expression of cytokines (about one-fifth for TNF-α, half for IL-1β and slightly lower for IL-6) and NO. The results are also coherent with our earlier ex-vivo assays in which postprandial TRL isolated from individuals consuming OA-enriched olive pomace oil attenuated the secretion of TNF-α and IL-1β in LPS-stimulated TPH-1 macrophages [45]. An analogy anti-inflammatory action of OA was reported by Matumba et al. [46] in skeletal muscle cells of rats fed a high-fructose diet, although the effect of the triterpene on the reduction of the TNF-α protein level was more modest than that on its gene expression. Similarly, β-amyrin, a biosynthetic precursor of OA, has demonstrated the ability to inhibit the production of PGE, TNF-α and IL-6 in RAW 246.7 macrophages stimulated with LPS [47], and the production and release of TNF-α, IL-1β and IL-6 in mice [48]. Simão da Silva et al. [48] attributed the anti-inflammatory effect of β-amyrin to the activation of stress signalling pathways such as the MAPK cascade. Papyriogenin D, another oleanane-type triterpene, also exhibits potential to inhibit NO production and the COX-2, TNF-α, and IL-6 expression in LPS-stimulated BV-2 cells [49].

We observed discrepancies between gene and protein expression, particularly for IL-1β when treated with TRL-OA. One would expect that reduced mRNA expression would lead to reduced protein expression, but this is not always the case. Vogel and Marcotte [50] showed that the correlation between mRNA and protein expression is relatively poor, which is likely due to post-transcriptional regulation and measurement noise. Subsequently, Perl et al. [51] tested this hypothesis using data from the transcriptome and proteome of the mouse auditory system. They found that protein levels are more conserved than mRNA levels in all tissues except cancer, which was attributed to buffering between transcription and translation. This phenomenon ensures that proteins can be made rapidly in response to a stimulus.

In the case of AT, its inclusion in lab-made TRL diminished the expression of genes encoding for IL1-β, IL-6, TNF-α and iNOS, although these actions were only reflected in the reduction of IL-6 level. By contrast, the presence of BS in TRL did not modify cytokine levels in BV-2 cells, despite the decrease of IL-1β gene expression. We believe that the contribution of AT to the attenuation of IL-6 release was not enough to have a significant effect on the mixture of the three compounds. However, treatment with BS was very effective in inhibiting NO release, likely through a post-translational process, since iNOS expression was not affected by this phytosterol. This behaviour diverges slightly from that documented for fucosterol, an antidiabetic phytosterol isolated from algae, which was able to significantly inhibit both the NO production and iNOS expression in LPS-stimulated RAW264.7 macrophages [52]. Finally, when we included the three biomolecules in the lab-made TRL composition, the obtained effects were, in general, the additive contribution of the individual components.

Our results also show that lab-made TRL cause oxidative stress that microglia compensate for by activating its endogenous antioxidant system. This work proposes that the inclusion of antioxidant biomolecules in the TRL core might help BV-2 cells in this task. Indeed, the presence of OA in lab-made TRL restored the GSH level to that in unstressed cells. This result is consistent with that found in our previous research, in which pre-treatment of BV-2 cells with OA restored GSH levels that were altered by exposure to LPS [13]. The inclusion of AT and BS in the TRL matrix, however, did not bring back significantly GSH concentrations, although a slight decrease could be observed. This fact might reflect that the antioxidant activity of AT and BS might operate through different molecular mechanisms than that of OA. OA is a potent activator of Nrf2, which stimulates the transcription of phase II response genes, including genes involved in GSH biosynthesis (glutamate-cysteine ligase and glutathione synthase) and regeneration (glutathione reductase) [53].

Our study shows for the first time that BV-2 microglial cells can be stimulated by lab-made TRL and that they can take up these particles into the cells, eliciting important effects on their inflammatory response. However, the mechanisms associated with the incorporation of TRL and their lipophilic content within the microglial cell and the triggering of the inflammatory response are not well established and should be subject to future investigations. Nevertheless, we can speculate that these processes might be somehow similar to those taking part in periphery macrophages, in which artificial TRL lead to the release of inflammatory markers after being incorporated by both receptor-dependent and independent pathways [54]. The receptor-related pathways include the LDL receptor, the LDL-receptor-related protein-1 (*LRP-1*) and the VLDL receptor, which are dependent on the presence of ApoE in the particles, but also include the scavenger receptor class B type 1 (*SR-B1*) and the scavenger receptor class A type 2 (*SR-A2*), as well as CD36. Once internalized, artificial TRL are capable of triggering the expression and release of proinflammatory cytokine and chemokines [55,56] as well as ROS generation [55], likely by means of NF-kB activation [57].

Microglial cells are present with *LDLR* and *LRP-1* receptors [58], and there is evidence that *LRP-1* is crucial for the reversion of microglial activation through the suppression of the NF–κB and MAPK signalling [59] and M1/M2 polarization [60]. On the other hand, *LDLR* overexpression in microglia is associated with suppressed microglial activation [61]. Moreover, mice lacking LDR showed enhanced amyloid deposition [62], reduced cognitive function, increased blood–brain barrier transport, and inflammatory responses when fed a Western diet [63]. Scavenger receptors might also be involved in TRL uptake by microglia as *SR-A2* expression is increased upon stimulation with LPS [64], and amyloid fibril Aβ and PrP increase mRNA expression in BV-2 cells [65]. Likewise, as we aforementioned, the uptake of TRL by microglial cells and the inflammatory response could also be mediated by the ApoB48R and its transcriptional activity [15].

Although the TRL used in the present investigation were prepared in the lab, and our results should be corroborated in the next studies with human TRL, they allow draw out important physiological implications from both the pharmacological and dietary points of view. The inclusion of OA in the lipophilic core of the TRL has demonstrated a higher efficiency in modulating the activated phenotype of BV-2 microglial cells.

In the last two decades, OA has been subject to intensive research. A number of dietary sources of these biomolecules and multiple beneficial effects in the prevention of chronic disorders in which oxidative stress and inflammation have been described [66]. The anti-inflammatory activity of this triterpene in different pathophysiological scenarios is well documented [53]. Many inflammatory cytokines are regulated by the transcription factor nuclear factor (NF)-kB, which is activated by endogenous and exogenous stimuli via the phosphorylation of the inhibitory subunit IkB by the IKK kinase complex. OA inhibits IKK, the core element of the NF–κB cascade. Furthermore, OA is a potent enhancer of the antioxidant adaptive cell response by its ability to transactivate Nrf-2. As such, OA increases the transcription of antioxidant enzymes, as well as genes involved in GSH biosynthesis and regeneration. As OA, but also AT and BS, can be found in virgin and pomace olive oils, they can be ascribed as parts of the Mediterranean diet [67]. In recent years randomized controlled trials with OA-enriched olive oil have being performed to benefit of the antioxidant and anti-inflammatory activities of the triterpene in prediabetic patients [68], and analogous trials should be addressed to research the therapeutic use of OA in AD patients.

In summary, this work offers new data supporting that TRL stimulates the inflammatory response in BV-2 microglial cells. The BV-2 cell line is widely used to study microglial function in vitro as a convenient model since mice’s primary microglia is limited in terms of the number of cells that can be obtained and is time-consuming [69]. In this regard, Henn et al. [70] compared the BV2 cells and primary cultures in response to LPS, finding that 90% of genes were induced both by tBV2 cells and primary microglia, although the latter to a lower extent. Immortalized human microglia (HMO6) are also available but more infrequently used as it is derived from human embryos, with ethical and legal issues. Additionally, HMO6 is defective for the release of lL-1β and NO when activated [71].

We also demonstrated here that the presence of bioactive lipophilic compounds in the lipoprotein matrix may attenuate the oxidative and chemotoxic stress associated with inflammation. The different effects observed with the assayed biomolecules could be due to the diverse pathways prompted by them in the modulation of the release of proinflammatory mediators by microglial cells. We cannot rule out that these biocompounds might modify TRL metabolism in microglia, since recent studies have described significant changes in microglial expression of genes involved in lipid and lipoprotein metabolism in response to deleterious signals in the brain [72]. The knowledge provided in this study adds evidence to the importance of lipophilic bioactive compounds as modulators of oxidative stress and the microglial inflammatory response. Moreover, our results open the way to the exploitation of their neuro-pharmacological potential through their delivery to the brain as part of TRL.

## 4. Materials and Methods

### 4.1. Oleanolic Acid Isolation and Characterization

High purity OA was obtained from olive tree leaves according to the procedure described by Albi et al. [73]. Briefly, olive tree leaves were extracted by maceration with 96% ethanol (20 mL/g leaf) at room temperature. The ethanolic extract was obtained by filtration and vacuum concentrated on inducing OA crystallization. OA crystals were separated from the concentrate by filtration and washed with cold 96% ethanol (5–7 °C), for the elimination of pigment traces and other possible contaminants. Finally, the crystals were submitted to a heat treatment at 165 °C and homogenized to a powder. OA purity was determined by gas chromatography [74].

An amount of 100 μL of a methanolic solution of betulininc acid (0.5 mg/mL) was added as an internal standard to an aliquot (100 μL) of the purified OA in methanol. The mixture was evaporated to dryness under a N2 stream, and the residue immediately dissolved in 200 μL of the silylating reagent (BSTFA+1%TMCS in pyridine) and incubated for 25 min at room temperature. OA identification was firstly carried out using a coupled gas chromatography-mass spectrometry detector (GC-MS) QP2010 Ultra (Shimadzu Europa GmbH) fitted with an AOC-20i autosampler, an ion source of electron impact and a quadrupole detector. The splitless mode was used and the injector temperature was set at 290 °C. Helium as a carrier gas at a pressure of 53.1 kPa and a flow of 1 mL/min was used. The oven temperature program was as follows: initial temperature, 50 °C for 1 min; 50–200 °C at 40 °C/min; 200–280 °C at 10 °C/min; and finally held for 2 min. Total run time was 14.75 min. The MS conditions were interface temperature, 280 °C; ion source temperature, 220 °C; electron impact, 70 eV; acquisition mode, scan (*m*/*z* 50–600). The identification of OA was accomplished by comparing the retention times and abundance ratios of two fragments of ions (203 and 189 *m*/*z*).

For OA quantification, 1 µL of the silylated sample was injected in an Agilent 6890 N GC (Agilent Technologies, Santa Clara, CA, USA), equipped with a Rtx-65TG Crossbond capillary column (30 m × 0.25 mm i.d.; 0.1 mm film thickness) coated with 35% dimethyl and 65% diphenyl polysiloxane as stationary phase (Restek, Co., Bellefonte, PA, USA) and an FID detector. The injection was realized in split mode, and hydrogen was used as carrier gas (pressure at column head 140 kPa). The oven temperature was isothermally established at 260 °C for 10 min. The injector and detector temperatures were established at 300 °C. The retention time for OA under these chromatographic conditions was 7.7 min.

### 4.2. Preparation of Lab-Made TRL

Lab-made TRL were prepared as described by Antelo and Perona [25]. Briefly, triolein (70%), cholesterol (2%), cholesteryl ester (5%) and phospholipids (25%) (all from Sigma-Aldrich, St. Louis, MO, USA) were mixed in 0.9% NaCl in tricine buffer (20 mmol/L, pH 7.4). The phospholipid composition was phosphatidylcholine (70.50%), lysophosphatidylcholine (6.88%), phosphatidylethanolamine (11.00%), phosphatidylinositol (2.58%), phosphatidylserine (2.58%) and sphingomyelin (6.54%) (Sigma-Aldrich). TRL-OA, TRL-AT, TRL-BS were synthesized by adding OA, AT and BS at a concentration of 10 µM, according to the highest non-toxic effective concentration observed in a previous study18. In addition, TRL containing a mixture of the three compounds at the same concentration was prepared to study possible synergies (TRL-M). AT and BS were purchased from Larodan (Larodan Inc., Malmö, Sweden). Lipids were sonicated by immersing an ultrasonic probe (Bandelin Electronics, Berlin, Germany) at 50 W for 20 min at 56 °C. The resulting emulsion was brought to a density of 1.21 g/mL with KBr, layered under a density gradient and centrifuged at 17000 g for 20 min at 20 °C (Beckman Optima L-90K centrifuge; Beckman Coulter, Palo Alto, CA, USA) in a SW41Ti swing-out rotor. The upper layer was removed and replaced with an equal volume of NaCl solution, and tubes were centrifuged at 70,000× *g* for 1 h (20 °C) in the same rotor. The upper layer (60 < Sf < 400) was collected and stored under N2 at −20 °C.

### 4.3. Microglia Cell Model

BV-2 brain microglia cells (Available online: http://bioinformatics.hsanmartino.it/cldb/cl7130.html (accessed on 2 June 2022)) were used as a microglial cell model. BV-2 (CVCL_0182) is a type of microglial cell derived from C57/BL6 murine that have been immortalized by v-raf/v-myc carrying J2 retrovirus. These cells have morphological, phenotypic and functional markers of macrophages. The cells were kindly provided by Alberto Pascual (Institute of Biomedicine of Seville, IBIS, Seville, Spain), and cultured in DMEM medium (Biowest, Nuaillé, France) supplemented with 10% heat-inactivated fetal bovine serum (FBS) (Biowest, Nuaillé, France) and antibiotics (100 U/mL penicillin and 100 U/mL streptomycin) (Biowest, Nuaillé, France), under a 100% humidified atmosphere of air + 5% CO_2_ at 37 °C. Cells were passaged every 2–3 days to maintain growth. The experiments assessed the effect of the prepared TRL with and without bioactive compounds. The volume of TRL was added according to its triglyceride concentration (0.15 μmol/mL culture medium), and saline (0.9%, pH 7.4) was used as a reference for calculations. Microglial phenotype and gene expression were carried out in a medium reduced to 5% of FBS.

### 4.4. Oil Red O Staining

BV2 cells were seeded at 5 × 10^5^ cells/mL and treated with TRL for 24 h at 37 °C. After incubation, cells were washed with PBS and treated with isopropanol 60% for 2 min. Isopropanol was removed and Oil Red O (0.2% (*m*/*v*) in 40% isopropanol/water (*v*/*v*)) was added and incubated for 10 min. Cells were washed with PBS and glycerol (30% in water, *v*/*v*) was added for preservation. A Motic AE21 Series (Motic, Barcelona, Spain) microscope coupled to a Moticam 2500 5.0 M Pixel Live Resolution camera (Motic, Barcelona, Spain) was used for microphotography.

### 4.5. Inflammatory Cytokine Production

BV2 cells were seeded at 5 × 10^5^ cells/mL in 6-well plates and treated with the different prepared TRL at 10 µM for 24 h at 37 °C. Culture media were collected, and the IL-6, IL-1β and TNF-α production was measured using ELISA kits (Diaclone, Besancon, France) according to the manufacturer’s instructions. Absorbance was measured at 450 nm using a scanning multiwell spectrophotometer (Multiskan spectrum, Thermo Fisher Scientific, Waltham, MA, USA).

### 4.6. Nitric Oxide Assay

BV2 cells were seeded in a 96-well plate at a density of 5 × 10^5^ cells per well and incubated overnight. Cells were treated with TRL, TRL-OA, TRL-AT, TRL-BS or TRL-M at the aforementioned concentration for 24 h at 37 °C. After the treatments, culture supernatants were collected, and the NO concentration was measured using a Griess reagent assay kit (R&D Systems, Inc. Minneapolis, MN, USA), which measures the levels of accumulated nitrite, a NO metabolite. Absorbance at 540 nm was determined in the Multiskan spectrum spectrophotometer.

### 4.7. Glutathione Assay

The culture supernatants after TRL treatments were also collected to analyze the total glutathione level by using a modification of the glutathione reductase recycling assay [75]. In this assay, reduced glutathione (GSH) is oxidized by 5,5′-dithiobis-(2-nitrobenzoic acid) (DTNB), resulting in the formation of oxidized glutathione (GSSG) and 5-thio-2-nitrobenzoic acid (TNB). GSSG is then reduced to GSH by glutathione reductase (GR) using reducing equivalents provided by NADPH. The rate of TNB formation is proportional to the sum of GSH and GSSG present in the cell culture sample and was determined at 412 nm with the Multiskan multiwell scanning spectrophotometer.

### 4.8. Gene Expression

Gene expression was measured by reverse transcription-quantitative polymerase chain reaction (RT-qPCR) by means of a CFX96 Real-Time PCR system (Bio-Rad, Hercules, CA, USA). BV2 cells were seeded at 5 × 10^5^ cells/mL in six-well plates and treated with TRL, TRL-OA, TRL-AT, TRL-BS or TRL-M at the aforementioned concentration for 24 h at 37 °C. Total RNA was isolated using Trisure (Bioline, Meridian Life Science, Inc. Memphis, TN, USA) and reverse transcribed to cDNA using a synthesis kit (NZYTech, Lisboa, Portugal). Quantitative PCR was performed by means of a Precision Plus Mastermix kit containing SYBR Green (PrimerDesign Ltd., Chandler’s Ford, UK), according to the manufacturer’s instructions and the following conditions: Denaturation at 95 °C for 2 min, followed by 40 cycles at 95 °C for 15 s and at 60 °C for 60 s. The relative gene expression was quantified according to the 2-ΔΔCt method, and the results were expressed as fold difference with the control (TRL) after normalization with hypoxanthine phosphoribosyl-transferase (HPRT). The following mouse primer sequences were used: IL-1β, forward GACCTTCCAGGATGAGGACA, reverse AGCTCATATGGGTCCGACAG; IL-6, forward AGTTGCCTTCTTGGGACTGA, reverse TCCACGATTTCCCAGAGAAC; TNF-α, forward AGTCCGGGCAGGTCTACTTT reverse GAGTTGGACCCTGAGCCATA; iNOS, forward CTCACTGGGACAGCACAGAA, reverse GGTCAAACTCTTGGGGTTCA; and HPRT, forward TGCTCGAGATGTCATGAAGG and reverse TATGTCCCCCGTTGACTGAT.

### 4.9. Statistical Analysis

All experiments were performed in triplicate, and the data was expressed as mean ± standard deviation (SD) of three independent experiments and analysed using the IBM SPSS Statistics 23.0 (IBM Corp., Armonk, NY, USA) software. Comparisons between the control and treatment groups were assessed by one-way ANOVA, followed by Tukey’s post-hoc test. The significant difference was set at *p* < 0.05.

## Figures and Tables

**Figure 1 ijms-23-07706-f001:**
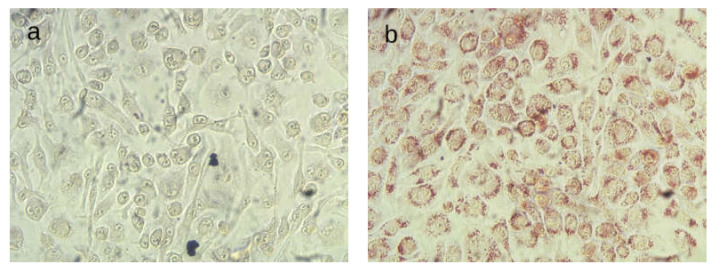
Oil Red O microphotographs of untreated (**a**) BV-2 microglial cells or treated with TRL (**b**) for 24 h at 37 °C.

**Figure 2 ijms-23-07706-f002:**
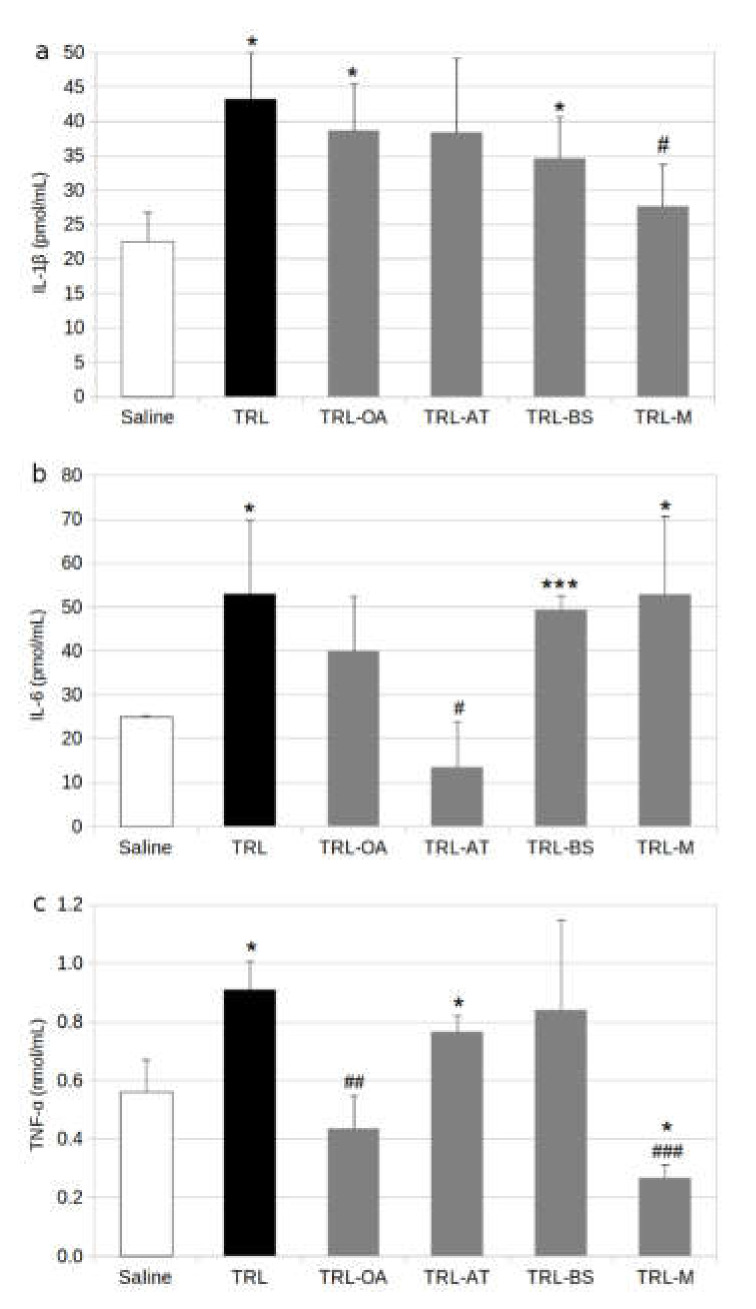
Cytokine release in BV-2 cells stimulated by TRL (positive control) or TRL carrying oleanolic acid (TRL-OA), α-tocopherol (TRL-AT), β-sitosterol (TRL-BS) or a mixture of the three compounds (TRL-M) for 24 h: (**a**) Interleukin-1 β (IL-1β); (**b**) Interleukin-6 (IL-6); (**c**) Tumour necrosis factor-α (TNF-α); Saline (0.9%, pH 7.4) was used as references for calculations. Values are expressed as mean ± SD of three independent experiments. Different signs indicate significant difference (* *p* < 0.05, *** *p* < 0.001 vs. saline; # *p* < 0.05 vs. TRL, ## *p* < 0.01 vs. TRL, ### *p* < 0.001 vs. TRL) by one-way ANOVA analysis and Tukey’s post-hoc test.

**Figure 3 ijms-23-07706-f003:**
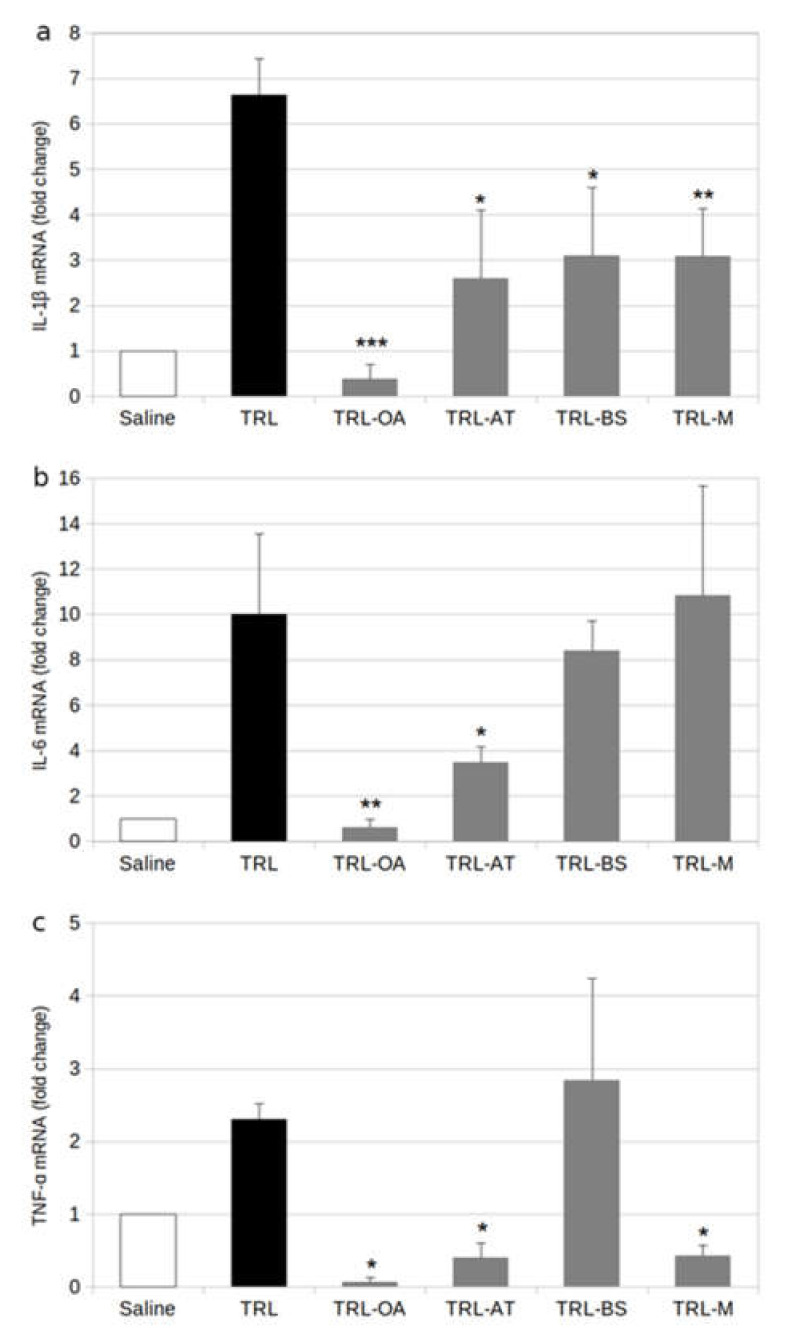
Cytokine RNA expression of cytokines in BV-2 cells stimulated by TRL (positive control) or TRL carrying oleanolic acid (TRL-OA), α-tocopherol (TRL-AT), β-sitosterol (TRL-BS), or a mixture of the three compounds (TRL-M) for 24 h: (**a**) Interleukin-1 β (IL-1β); (**b**) Interleukin-6 (IL-6); (**c**) Tumor necrosis factor-α (TNF-α); Saline (0.9%, pH 7.4) was used as references for calculations. Values are expressed as mean ± SD of three independent experiments. Different signs indicate significant difference (* *p* < 0.05, ** *p* < 0.01, *** *p* < 0.001 vs. saline) by one-way ANOVA analysis and Tukey’s post-hoc test.

**Figure 4 ijms-23-07706-f004:**
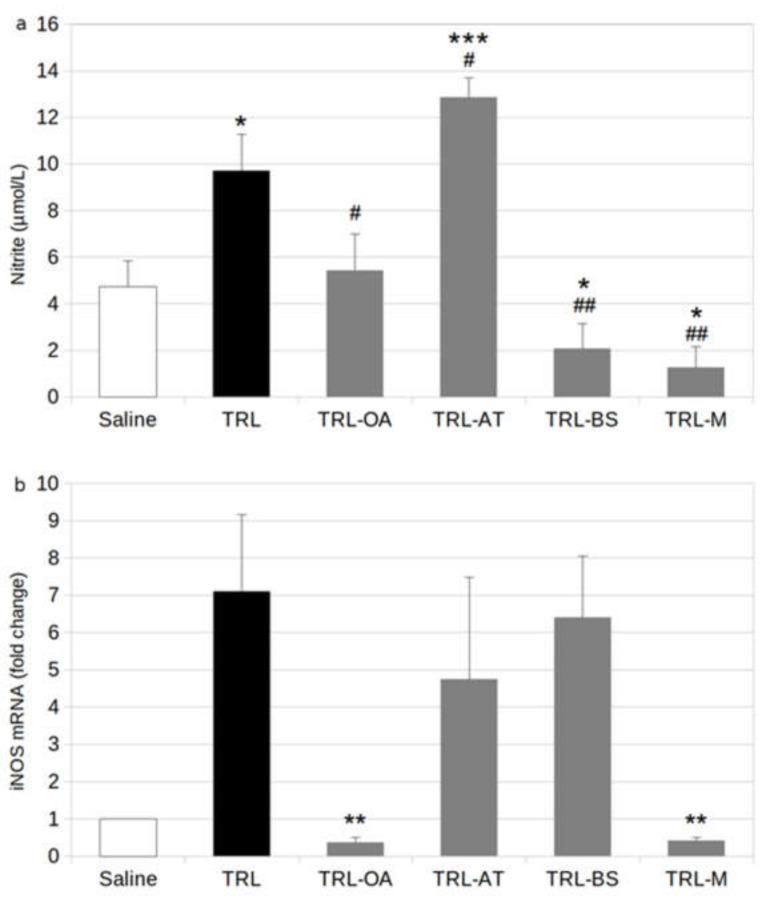
(**a**) Nitrite production and inducible nitric oxide synthase; (**b**) (iNOS) gene expression in BV-2 cells stimulated by TRL or TRL carrying oleanolic acid (TRL-OA), α-tocopherol (TRL-AT), β-sitosterol (TRL-BS) or a mixture of the three compounds (TRL-M) for 24 h, Saline (0.9%, pH 7.4) was used as references for calculations. Values are expressed as mean ± SD of three independent experiments. Different signs indicate significant difference (* *p* < 0.05, ** *p* < 0.01, *** *p* < 0.001 vs. saline; # *p* < 0.05, ## *p* < 0.01, vs. TRL) by one-way ANOVA analysis and Tukey’s post-hoc test.

**Figure 5 ijms-23-07706-f005:**
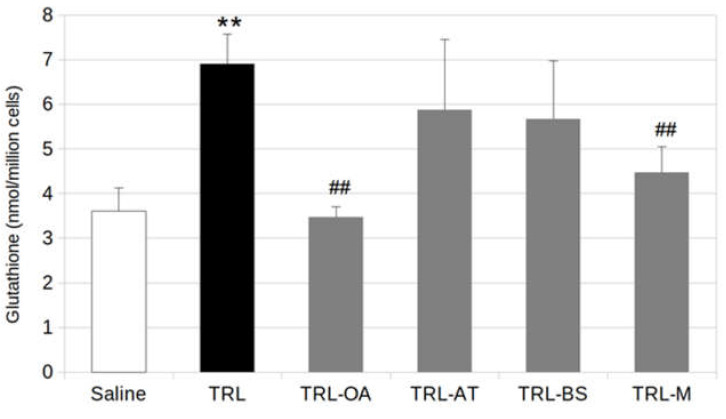
Intracellular concentrations of total glutathione (GSH + GSSG) in BV-2 cells stimulated by TRL or TRL carrying oleanolic acid (TRL-OA), α-tocopherol (TRL-AT), β-sitosterol (TRL-BS) or a mixture of the three compounds (TRL-M) for 24 h, Saline (0.9%, pH 7.4) was used as references for calculations Values are expressed as mean ± SD of three independent experiments. Different signs indicate significant difference (** *p* < 0.01, vs. saline; ## *p* < 0.01, vs. TRL) by one-way ANOVA analysis and Tukey’s post-hoc test.

## Data Availability

Publicly available datasets were analysed in this study. This data can be found at the Digital CSIC Repository. Available online: https://digital.csic.es/ (accessed on 2 June 2022).

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
