# Peer review of "Lipophilic Bioactive Compounds Transported in Triglyceride-Rich Lipoproteins Modulate Microglial Inflammatory Response"

_ijms, 2022, doi:10.3390/ijms23147706_

Round 1
Reviewer 1 Report
Summary: Dietary fats are transported in the blood around the body as triglyceride-rich lipoproteins (TRL). TRL are associated with atherosclerosis, and recently certain lipids have been associated with the pathogenesis of Alzheimer’s disease (AD). This paper assesses the effect of TRL on the pro-inflammatory activation of a microglial (brain inflammatory cells) cell line in vitro. The cell line used is a microglial/macrophage-like, mouse-derived immortalised cell line BV2. TRL was made in house in the laboratory. The authors find that 10uM TRL causes significant increases in three major pro-inflammatory microglial cytokines IL1b, IL6 and TNFa at both RNA (by qRT-PCR) and protein (by ELISA) level. Microglial nitrate and iNOS (a marker of nitric oxide synthesis) are also increased by TRL, and glutathione is increased (marker of antioxidant activity). The authors then tested incorporation of specific lipophilic bioactive compounds oleanolic acid, a-tocopherol and b-sitosterol, commonly found in dietary oils, into the TRL, in the same assays. These variably affected the inflammatory/activation markers, with a number of reductions in the TRL-induced increase of activation. The most significant effect is showing that TRL has large pro-inflammatory effects on microglia. Secondly, oleanolic acid has significant and consistent effects on a number of measures. The compound mix (TRL-M) also shows some significant reductions on inflammatory markers.
Strengths of the study: The manuscript is written clearly for the most part and is well-structured. The concept is useful. A number of major and appropriate markers are analysed. Graphs are complete. Design of experiments is complete, appropriate and seems reasonable. Given methodology is clear and appropriate. Statistics and biological/technical replicates are good and appropriate. Lab-generation of TRL is a good idea and a useful method for future use. Results are quite well discussed in the Discussion with relevant literature compared. There is little on TRL and microglia in published literature, so this is a good study to add to current knowledge.
Main areas of weakness: In my view, aside from the TRL and oleanolic results mentioned above, the other results are more varied and more difficult to make consistent conclusions from. The hypotheses, and the significance and practical implication of the results, are not described clearly enough. The significance of TRL-carriers inducing a large pro-inflammatory response needs to be discussed, in the context of a pharmacological approach. I would like to see data of effects on microglia cell proliferation/number and viability. Some discussion or investigation of the mechanism of the TRL-lipoprotein effects would be warranted.
Specific comments
The manuscript is well written, but there are a few English language inaccuracies/unclear words (see some examples below).
Line 16 ‘reduced’ : The conclusion needs to be more specific (i.e. which biomolecules reduced which measures). Some of the bioactive molecules only reduced certain cytokines/NO or inconsistently.
Line 39 Specify what impact on TRL-Ab.
Line 45 It would be useful to state glutathione as an example of the antioxidant system, for non-specialists.
Line 61/62 It would be useful to briefly describe the cell identity and origin of BV-2 cells.
Line 67 State that this was tested by ELISA, in order to differentiate from the qRT-PCR used in section 2.2
Line 95 State that this was tested by qRT-PCR, in order to differentiate from the ELISA used in section 2.3
Line 117 mRNA names e.g. IGF-1 should be given italicised throughout the manuscript, to distinguish from protein.
Line 144 iNOS gene ‘expression’
Line 147 Please state what GSH is measuring in this section (antioxidant system).
Line 172 It would be useful to state if the secretion of inflammatory cytokines in references 31 and 32 are specific to astrocytes or also include microglia.
Line 203 ‘emphasising’ – should this be ‘although’ or a similar term?
Line 251 ‘obtaining’ – replace with ‘isolation’ or a similar term.
Line 302 4.3 title – typo?
Line 305 ‘gently’ – kindly
Line 383 onwards. References #3-6 and 26 are around twenty years old; it would be better to replace these with newer reviews/articles.
Introduction:
Hypotheses should be drawn out more clearly (in abstract and around line 60-62).
Results :
Specific the volume/% of saline added to control samples.
Comparison of the degree of activation by TRL to LPS in these cells would be useful, as LPS is commonly used as a standard activator and used previously by the authors.
I would like to see effects on microglia cell proliferation/number and viability – is this affected by TRL particularly, but also by the TRL-biomolecules? It would be useful to include images of cells, including the activated phenotype if it is present.
Discussion :
The significance of the results needs to be made more clear. For non-specialists, more attention to the practical implications and significance of TRL-carriers as a potential neuropharmacological approach is needed.
The significant of TRL-lipoproteins inducing a large pro-inflammatory response should be discussed further. For example, the OA-carrying TRL shows significant activation compared to saline control, so there is clearly still activation induced. How would this translate to dietary or neuropharmacological interventions? Would the compounds be intended to decrease microglial activation that is already present in the brain?
Some discussion that BV2 is a mouse cell line – and how comparable it may be to human microglia in vivo – would be warranted.
Some discussion of the mechanism by which TRL and TRL-biomolecules are activating/attenuating activation of MG should be given. E.g. Is there evidence that the MG take up the TRL or are extracellularly activated by TRL? What specific intracellular pathways are/might be functioning from TRL stimulation to then cause MG activation? What sort of transporters are present in BV2 cells and are these expression patterns similar to human microglia? This could be analysed by RT-PCR or using online publicly available expression datasets.
A conclusion of which specific TRL-biomolecules would be best to continue using in future research would be useful. Some discussion of how exactly this would be taken into humans would be useful. In the future this would be great to test in a mouse AD model or human cell line AD model.
Author Response
On behalf of all the co-authors, we would like to thank the reviewer for their comments, which have undoubtedly contributed to the substantial improvement of the manuscript.

Reviewer 2 Report
This article, entitled “Lipophilic bioactive comopounds transported in triglyceride-rich lipoproteins modulate microglial inflammatory response,” suggested a decrease in microglial activation using lab-made triglyceride-rich lipoproteins (TRL). However, data to support this proposal are severely lacking.
Author Response

(The authors gave the same response as above.)

Reviewer 3 Report
In this work, the authors assess the role of triglyceride-rich lipoproteins loaded with different bioactive compounds on microglia activation using the well-studied BV-2 microglia cell line.
The experimental design is clear and straightforward, and the results support the authors’ conclusions.
Major issues:
The statistical analyses performed to analyze each figure must be reported in more detail, in the figure legends. For example, it is imperative to describe if the data were first analyzed by one- or two-way ANOVA and what were the results of said analyses before one can proceed with posthoc analyses.
· How do the authors explain the results in Fig. 1b indicating that TRL loaded with all three compounds had no effects on IL-6 levels when TRL loaded with AT alone did? How does the presence of BS and OA block the increase in IL-6 gene expression?
· Have the authors controlled for cell viability after each subministration of empty or loaded TRLs?
· The results in Fig. 1a and 2a are counterintuitive. Fig. 1A shows that the protein levels of IL1β were not reduced by TRL-OA and yet gene expression of IL1β was drastically reduced by TRL-OA (Fig. 2a). This apparent discrepancy must be explained by the authors.
· The same criticism raised above is valid when comparing protein and mRNA levels of IL1β after TRL-BS and TRL-AT treatment.
Author Response

(The authors gave the same response as above.)

Round 2
Reviewer 1 Report
All revisions requested have been completed to a high standard. This is interesting work which is presented well.
Reviewer 3 Report
The authors have successfully addressed my comments.